

# Hybrid contour and geometric partitioning for accurate plantar foot region segmentation

Shumei Zhang[1,2], Xi Liang[3], Minmin Wu[1,2] and Weiming Gu[3]

[1] College of Computer and Data Science, Putian University, Putian, Fujian, China
[2] Shoe and Clothing Industry Research Institute, Putian University, Putian, Fujian, China
[3] Custom Factory, Shuangchi Technology Co., Ltd, Putian, Fujian, China

## ABSTRACT

**Background**. Precise segmentation of plantar foot regions is crucial for analyzing foot structure and pressure distribution, aiding in the diagnosis of pathologies and enabling preventive interventions. However, conventional segmentation approaches often struggle to accurately delineate key anatomical regions and detect their boundaries, particularly in the presence of foot abnormalities.

**Methods**. We created a dataset of plantar pressure images and proposed a hybrid algorithm that integrates edge contour detection techniques with dynamic geometric partitioning to address persistent challenges in plantar region segmentation. Our method first determines the lengths of the left and right feet using precise contour detection, then partitions the plantar surface into primary anatomical regions (forefoot, midfoot, and heel) based on standardized geometric proportions. Additionally, the methodology allows for finer subdivisions (*e.g.*, inner/outer forefoot) that adapt to the unique morphology of each foot. This algorithm accommodates five foot types, including normal, low arch, high arch, inward heel tilt, and outward heel tilt.

**Results**. A comparative evaluation of three edge detection methods revealed that the Canny algorithm, when combined with geometric partitioning, yielded superior performance. On a dataset of 200 plantar pressure footprints encompassing both normal and abnormal feet, this hybrid approach achieved Intersection over Union (IoU) and mean Average Precision (mAP) scores exceeding 0.90 across all segmented regions (forefoot, midfoot, and heel). Furthermore, the results indicate that the proposed hybrid algorithm performs comparably across both normal and abnormal foot types, with no significant differences observed.

**Conclusions**. Our synergistic integration of contour detection and geometric partitioning yields an efficient technique for segmenting plantar regions from static plantar pressure images. Validation on a diverse dataset shows that the proposed approach accurately distinguishes foot-specific regions across five different foot types, including both normal and pathological cases.

Corresponding author
Shumei Zhang,
smz_111@outlook.com

## INTRODUCTION

The human foot, often referred to as the "second heart" of the body, plays a vital role in maintaining balance, locomotion, and systemic circulation. Its structural integrity is crucial for overall health, making accurate assessment essential. Plantar pressure imaging technologies have emerged as effective non-invasive tools for diagnosing and managing various health conditions (*Arzehgar et al., 2025*). Recent advancements in wireless systems have enhanced the practicality of plantar pressure measurement (*Dhanashri & Uttam, 2018*), enabling diverse applications through methods such as footprint analysis (*Soames, 1985*), plantar pressure scanners (*Wang et al., 2019*), pressure plates (*Nuyts et al., 2022*), and pressure insoles (*Hsu et al., 2023*). Initially focused on clinical applications, such as diagnosing foot abnormalities (*Farahpour et al., 2016*) and predicting diabetic foot ulcers (*Das, Roy & Mishra, 2022*), these technologies now also support gait analysis (*Xie et al., 2023*), athlete training (*Li, 2022*), and footwear design, where comfort and functionality are essential for promoting healthy foot function and preventing injuries (*Hao et al., 2020*).

A critical aspect of plantar pressure analysis is accurately segmenting the foot into distinct functional regions. As defined in anatomical literature (*Pinney, 2023*), these regions consist of the forefoot, midfoot (arch), and hindfoot (heel). Each functional region has specific biomechanical roles: the forefoot governs propulsion, the midfoot absorbs shock, and the hindfoot bears load during motion. These definitions are grounded in anatomical and biomechanical principles, with standardized geometric proportions often used to delineate regions. For instance, *Cavanagh & Rodgers (1987)* proposed the arch index (AI), which divides the footprint into equal thirds (rearfoot, midfoot, forefoot) perpendicular to the foot axis, excluding the toes. The AI calculates the ratio of the midfoot area to the total footprint area, classifying foot types as high arch (AI $\leq$ 0.21), normal (0.21 <AI <0.26), or flat arch (AI $\geq$ 0.26). Similarly, *Costa, Coelho & Silva (2022)* developed a multi-stage segmentation pipeline to identify key plantar regions for diabetic neuropathy testing, combining HSV color space thresholding, Canny edge detection, and anatomical landmark analysis. Traditional image segmentation techniques, such as thresholding (*Jardim, António & Mora, 2023*), watershed segmentation (*Kornilov & Safonov, 2018*), edge detection (*Jing et al., 2022*) (implemented *via Sobel & Feldman (1968)*, *Canny (1986)*, or Laplacian (*Wang, 2007*)), and region-based segmentation (*Mazouzi & Guessoum, 2021*), are integral to these analyses. While computationally efficient, these methods often struggle with anatomical variability and low-contrast boundaries (*Vd & Subedha, 2019*).

In contrast, data-driven approaches leverage machine learning (ML) and deep learning (DL) to automate the segmentation and analysis of plantar pressure images. These methods offer higher accuracy and automated learning capabilities but are limited by interpretability, clinical adaptability, and reliance on large annotated datasets (*Zhang, Shen & Jiao, 2024*). For example, *Wang et al. (2020)* combined a fully convolutional network (FCN) with SegNet to enhance the segmentation of plantar pressure images for shoe-last design and footwear comfort optimization. Their experimental results demonstrated that this hybrid approach outperformed traditional methods in accuracy and boundary delineation. However, a key limitation of their method is its reliance on the manual division of the

plantar surface into ten anatomical regions during data preprocessing, which may introduce subjectivity and limit scalability. *Dindorf et al. (2025)* employed a U-Net model to segment the plantar surface into four regions: the hallux, metatarsal area 1, metatarsal areas 2–5, and the heel. While they achieved near-expert accuracy, they faced challenges in labeling ambiguous regions, particularly metatarsal area 1. Therefore, accurately identifying the foot contour, partitioning functional regions, and detecting boundaries remain fundamental challenges in plantar image segmentation. Contours outline the shape of an object, connecting edge points to emphasize structural information (*Cai et al., 2024*). However, achieving precise segmentation is complicated by anatomical variability, irregular foot geometries, and pathological deformities.

Despite these advancements, several key challenges persist in plantar region segmentation:

1. Identifying Specific Regions: Extracting specific plantar regions (forefoot, midfoot, and heel) is complicated by individual variations in foot shape and size, as well as the presence of abnormalities, leading to inconsistencies in anatomical structures.
2. Precise Boundary Detection: Accurately defining the boundaries of these regions is crucial for effective pressure distribution analysis. However, detecting these boundaries is challenging, especially when edges are incomplete or fragmented.

To address these challenges, this study aims to develop a hybrid algorithm that combines contour detection with geometric partitioning methods. This approach focuses on improving boundary delineation through an optimized edge detection algorithm and enhancing anatomical region identification using standardized geometric protocols.

## MATERIALS & METHODS

### Ethics approval

The experimental paradigm was approved by the Research Ethics Committee (REC) of New Engineering Industry College, Putian University (NO. 2023030206), in compliance with the Declaration of Helsinki. Written informed consent was waived by the REC.

### Device for foot pressure scanning

Plantar pressure images were collected using a non-invasive foot pressure scanner (Footbird), equipped with flexible pressure sensors, as shown in Fig. 1A. The device comprises a self-designed motherboard and a high-precision flexible thin-film array piezoresistive pressure sensor. The sensor array consists of 14,400 sensing units arranged in a $120 \times 120$ grid, with each unit capable of detecting pressures ranging from 0 to 255 kPa. The total effective acquisition area is $42 \times 42$ cm$^2$, enabling real-time visualization of plantar pressure distribution on a display screen, as illustrated in Fig. 1B. Scanned images were stored for further analysis.

### Participants and recruitment

The study participants consisted of 400 adult volunteers (228 males, 172 females; aged $\geq 18$ years) who visited the Shuangchi Customized Factory shoe store for footwear customization services. Inclusion criteria required participants to be adults with no prior foot-related

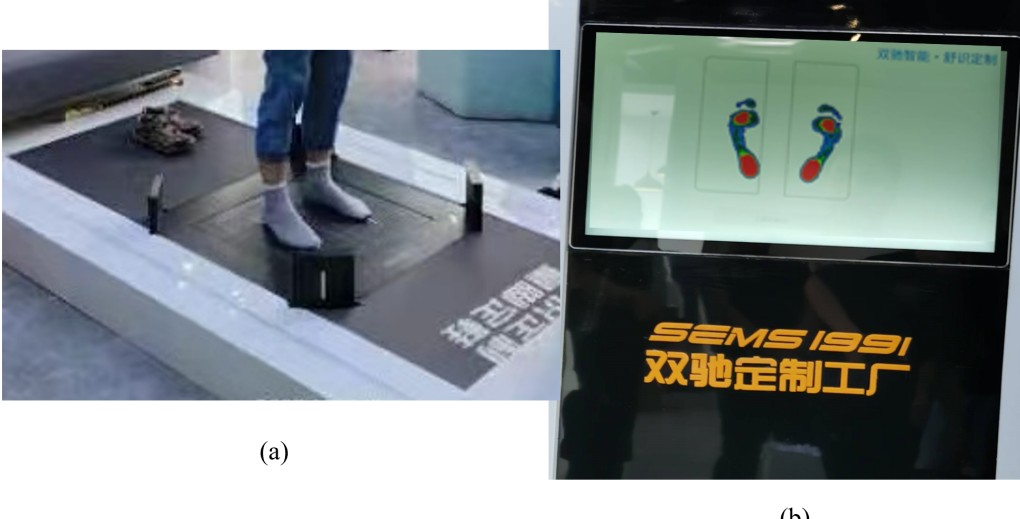

(a)

(b)

**Figure 1** **(A) The foot pressure scanner, named Footbird, (B) a sample plantar pressure image.**

medical conditions, while exclusion criteria excluded individuals with recent foot injuries or surgeries. Recruitment was conducted *via* in-store announcements, and participants were verbally informed of the study's purpose, with the option to decline participation. To prioritize privacy and focus on algorithmic evaluation, no personal identifiers (*e.g.*, age, height, weight) were collected.

## Data collection protocol

Before data collection, the Footbird system was calibrated to zero pressure. Participants removed their shoes, donned disposable socks, and stood motionless on the fixed foot area of the scanner for approximately 15 s until the plantar image appeared on the screen. The device captures 125 frames of grayscale pressure images, each measuring $700 \times 700$ pixels. These 125 images are then combined and processed into a single composite image, which is pseudo-colored according to the pressure values, as illustrated in Fig. 1B. Each volunteer contributed one footprint per foot, yielding a total of 800 plantar pressure foot samples from 400 participants.

## Sample size determination

To balance sample quality and quantity, we selected 200 footprints from the initial 800 plantar pressure images. This selection constitutes 25% of the total dataset, which is a common practice in many studies. The initial dataset included complete (full footprint), basically complete (missing some toes), and incomplete (missing partial foot regions) images, as demonstrated in Fig. 2. To ensure representativeness, selected images prioritized completeness (excluding severely fragmented scans) and diversity in foot morphology, including normal, low arch, high arch, inward heel tilt, and outward heel tilt, as shown in Fig. 3. The definitions of the five foot types are as follows (*Pinney, 2023*):

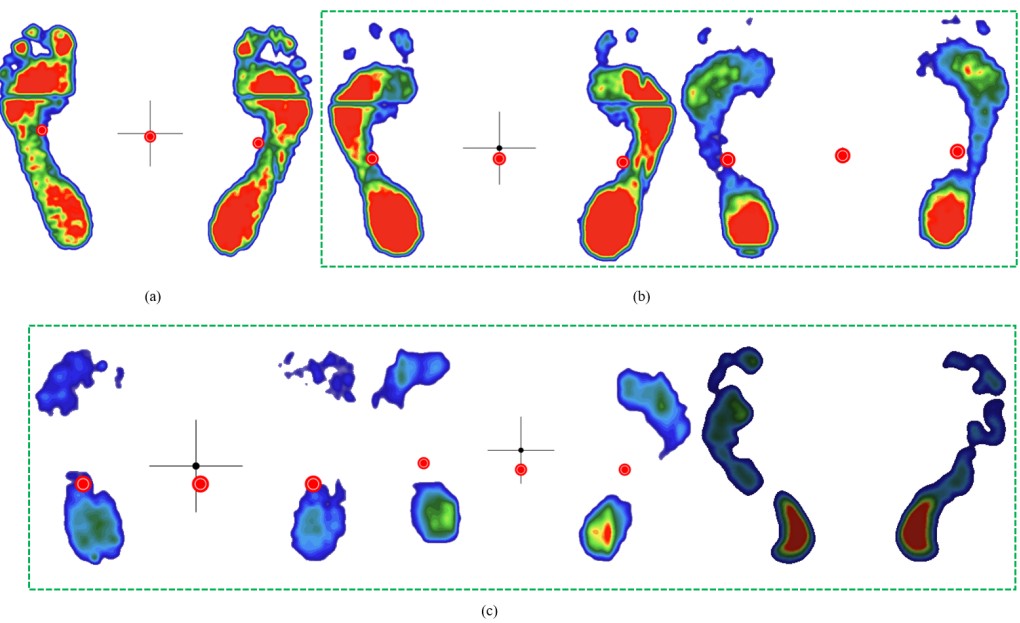

**Figure 2** **Examples of initially collected plantar pressure images.** (A) complete, (B) basically complete, (C) incomplete.

- **Normal foot**: Slightly curved midfoot (approximately 50% midfoot contact, Fig. 3A).
- **Low arch** (**flatfoot**): Near-complete midfoot contact (Fig. 3B).
- **High arch**: Minimal midfoot contact (Fig. 3C).
- **Inward heel tilt**: Calcaneus deviation inward; the toe area may exclude the hallux (Fig. 3D).
- **Outward heel tilt**: Calcaneus deviation outward; the toe area may show only the hallux (Fig. 3E).

## Plantar image segmentation algorithms

For the segmentation of plantar images, we evaluated three established algorithms: thresholding, region splitting and merging, and Canny edge detection. Each method has its advantages and disadvantages, and we focused on their effectiveness in segmenting the forefoot and heel regions.

### *Thresholding algorithm*

Thresholding is a widely used image segmentation technique due to its simplicity and ease of implementation. This method divides image pixels into different classes based on their gray levels by setting specific thresholds for target and background areas.

In this study, we employed simple thresholding, which involves setting a single threshold value applied uniformly to each pixel in the grayscale image. If the pixel value is less than the threshold, it is set to 0; otherwise, it is set to a specified maximum value (255).

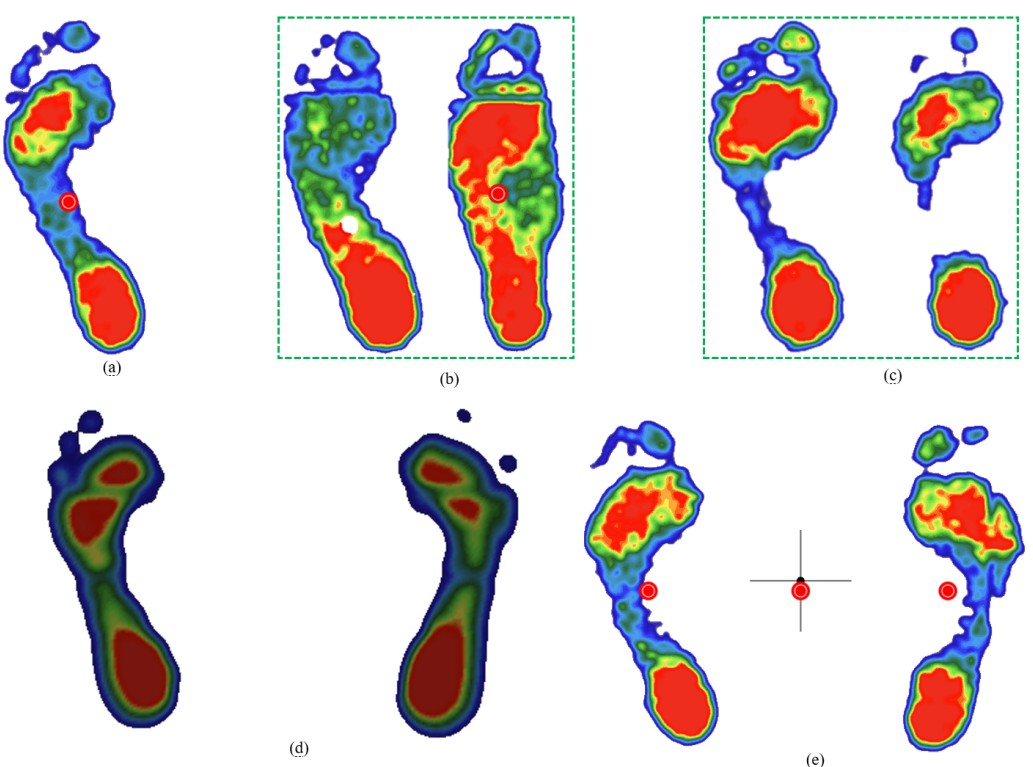

**Figure 3  Five foot types.** (A) normal, (B) low arche, (C) high arche, (D) inward heel tilt, (E) outward heel tilt.

The threshold value was set to 127 for this study, as shown in Eq. (1).

$$dst\,(x,y) = \begin{cases} maxVal, & if\ src\,(x,y) > thresh \\ 0, & otherwise \end{cases} \tag{1}$$

where $dst\,(x,y)$ is the pixel value in the binary image, $src\,(x,y)$ is the pixel value in the grayscale image, and maxVal is set to 255.

### Region split and merge algorithm

The region split and merge algorithm partitions an image into uniform regions. Initially, the image is divided into disjoint, internally coherent small regions. If only the splitting process is used, many neighboring regions may exhibit similar properties. To address this, a merging step is introduced, allowing adjacent regions to be merged based on homogeneity criteria. The following pseudo-code outlines the algorithm:

```
function regionSplitMerge(image):
# Initialize lists for processing and storing regions
processList = [image] # Start with the entire image as the only element
regionList = [] # List to store uniform regions
# Splitting Phase
while processList is not empty:
    region = Extract the first element of processList
# Check if the region is uniform based on the homogeneity criterion
if isHomogeneous(region):
        Add region to regionList # Store the uniform region
else:
    # Split the region into four equal sub-regions
    subRegions = divideRegion(region)
    Add subRegions to processList # Add sub-regions for further processing
# Merging Phase
minThresh = value1 # Define minimum threshold for merging
maxThresh = value2 # Define maximum threshold for merging
# Repeat merging until no more merges are possible
while true:
    merged = false # Flag to check if any merging occurred
    for each region in regionList:
        for each remainingRegion in regionList:
            # Check if regions are neighbors and meet the homogeneity criterion
            if areNeighbors(region, remainingRegion) and isHomogeneous(region, remain-
            ingRegion):
                # Merge the two regions
                mergedRegion = merge(region, remainingRegion)
                 Update regionList to replace region and remainingRegion with mergedRe-
                gion

                # Set pixel values based on intensity thresholds
                for each pixel in mergedRegion:
                    if pixelIntensity(pixel) is within (minThresh, maxThresh):
                    Set pixel as region boundary
                else:
                    Set pixel as background
            merged = true # Set flag to indicate a merge occurred
            break # Exit the inner loop to restart checking from the beginning
    if merged:
        break # Exit the outer loop to re-evaluate regions after a merge
```

It is evident from the pseudo-code that adjusting the merging thresholds (minThresh, maxThresh) can significantly impact the segmentation results.

### *Canny edge detection algorithm*

The Canny edge detection algorithm is a multi-stage process consisting of the following five steps:

- **Noise Reduction**: A Gaussian filter is applied to reduce noise.
- **Intensity Gradient Calculation**: The smoothed image is filtered to calculate the intensity gradient for each pixel.
- **Non-maximum Suppression**: This step removes unwanted pixels by checking if each pixel is a local maximum in its gradient direction.
- **Double Thresholding**: Two threshold values (minVal, maxVal) classify edge pixels into strong, weak, and non-edge categories.
- **Edge Tracking by Hysteresis**: Strong edges are identified while suppressing isolated weak edges.

Optimal double thresholds will be determined based on comparative experimental results.

## Contour extraction, filtering, and foot assignment

After applying the three segmentation algorithms (Thresholding, Region Split & Merge, Canny), the binary outputs underwent the following post-processing steps to extract and refine contours and assign them to the respective foot (left or right).

Contour detection identifies closed object shapes by locating continuous points along their boundaries with the same color or intensity. This process retrieves boundary lines from a binary image generated through the aforementioned segmentation techniques. Filtering enhances contours by removing smaller, less relevant regions, leading to clearer identification of the forefoot, midfoot, and heel regions.

In this study, contour detection employs the border-following algorithm (*Suzuki & Abe, 1985*). Each detected contour is stored as a vector of points in an array. A minimum area criterion is applied to filter out unwanted contours, calculated using Eq. (2), and the coordinates of the contour boundaries for regions of interest are stored in the intersContour array, as shown in Eq. (3).

$$area[i] = \frac{1}{2} \sum_{j=0}^{n_i} x_j y_{j+1} - x_{j+1} y_j \, | \, (x_j, y_j) \in contour[i] \qquad (2)$$

$$intersContour = intersContour + contour[i], if area[i] > minArea \qquad (3)$$

where $area[i]$ represents the area enclosed by the $i$th contour in the contour array; $n_i$ denotes the number of boundary points contained in the $i$th contour array; *minArea* is set to 6000 in this study.

To analyze the left and right feet separately, we assign regions of interest extracted from the entire image to each foot. This involves determining the minimum bounding rectangle (boundingRect) for each contour based on the intersContour array, as defined in Eq. (4).

                                                   

The relevant region information is then stored in the regions list, as shown in Eq. (5).

$$x, y, w, h = boundingRect(intersContour)$$
$$topLeft = (x, y); bottomRight = (x + w, y + h)$$
$$side = \begin{cases} left, & if \ x < imgWidth/2 \\ right, & otherwise \end{cases}.$$

(4)

$$regions = [(side_0, topLeft_0, bottomRight_0), \ldots, (side_m, topLeft_m, bottomRight_m)].$$

(5)

Here, $x$, $y$, $w$, and $h$ represent the coordinates of the top-left corner $(x, y)$ and the width and height of the contour's bounding rectangle. The variable *side* indicates whether the contour belongs to the left or right foot, and *imgWidth* is the width of the entire image.

## Combining contours and geometric segmentation for foot region analysis

To achieve precise segmentation of the forefoot, midfoot, and heel regions, we developed a hybrid approach that integrates the post-processed contour information with geometric partitioning techniques. This combined method proceeds as follows for the left foot (analogously for the right). Since abnormal feet may result in missing toe parts in plantar images, the proposed integrating algorithm uses an area threshold to exclude the toe region.

Initial calculations involve determining the minimum and maximum y-coordinate values for the left foot and the x-coordinate values for each region, as illustrated in Fig. 4. The foot length from forefoot to heel is determined based on the values in the contour regions, as shown in Eq. (6). This length is then divided into three regions: the forefoot, midfoot (arch), and heel, based on standard proportions (*Pinney, 2023*), as described in Eq. (7). The forefoot is further subdivided into outer and inner regions using Eqs. (8) and (9), while rectangular regions for the midfoot and heel are defined using Eqs. (10) and Eq. (11), respectively. Information about the four regions in the left foot is stored in the leftRegions list, as shown in Eq. (12).

$$leftL = \max(leftY) - \min(leftY) + 1$$

(6)

$$leftL\_fore, leftL\_arch, leftL\_heel = leftL * 0.3, leftL * 0.3, leftL * 0.4$$

(7)

$$outerFore = [(minF_x, \min(leftY)), (minF\_x + int\left(\frac{maxF_x - minF_x}{2}\right),$$
$$min(leftY) + leftL\_fore)]$$

(8)

$$innerFore = [(minF\_x + int\left(\frac{maxF_x - minF_x}{2}\right), min(leftY)), (maxF\_x,$$
$$min(leftY) + leftL\_fore)]$$

(9)

$$arch = [(minA\_x, \min(leftY) + leftL\_fore), (maxA\_x, \min(leftY)$$

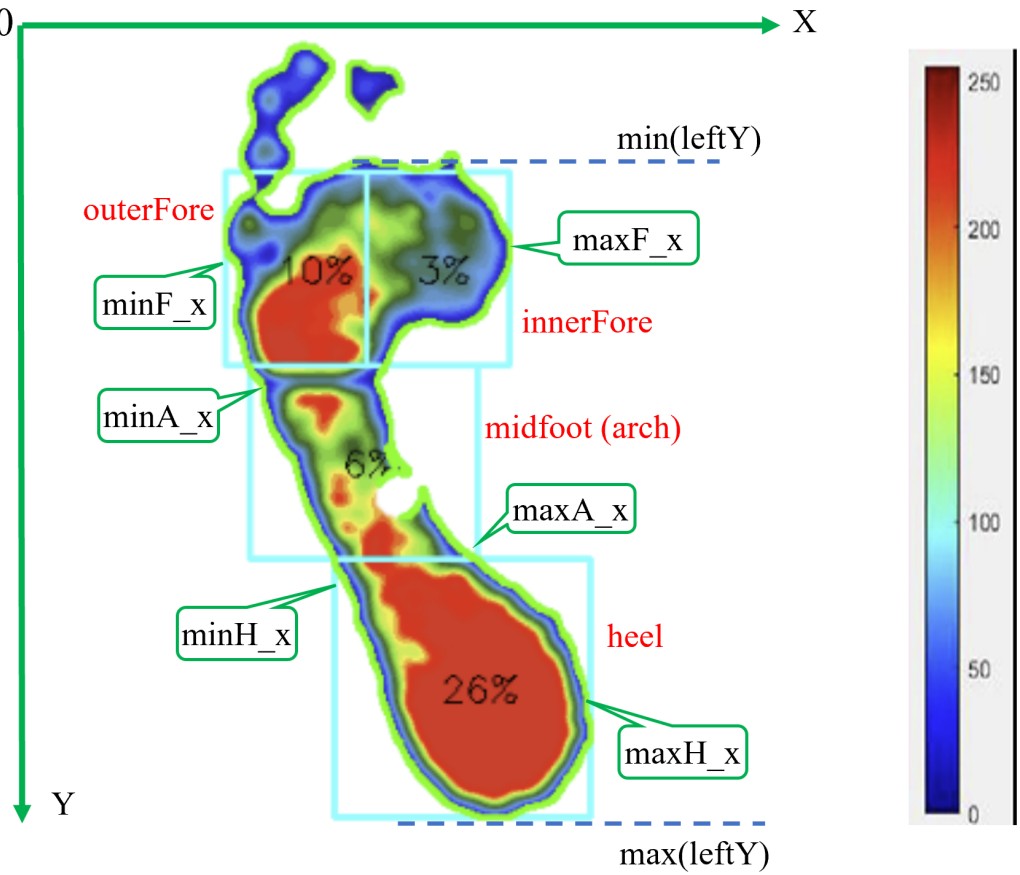

**Figure 4 Division of the left foot into four regions (outer forefoot, inner forefoot, arch, heel), with a color bar indicating pressure values measured in kPa for plantar pressure analysis.**

$$+ leftL\_fore + leftL\_arch)] \tag{10}$$

$$heel = [(\min H_x, \min(leftY) + leftL\_fore + leftL\_arch), (maxH\_x, \max(leftY))] \tag{11}$$

$$leftRegions = [ourerFore, innerFore, arch, heel] \tag{12}$$

where, min(leftY) represents the minimum $y$-coordinate of the left foot, and max(leftY) is the maximum $y$-coordinate of the left foot.

## RESULTS

To validate the accuracy and reliability of the plantar image segmentation algorithms, we utilized a test dataset consisting of 200 selected footprints from an initial pool of 800 plantar pressure images. This subset includes 80 footprints of normal feet, with the remaining 120 footprints comprising 30 footprints each of low arches, high arches, inward heel tilt, and outward heel tilt. This distribution ensures comprehensive coverage of both normal and abnormal conditions. The specific experimental processes and findings are detailed below.

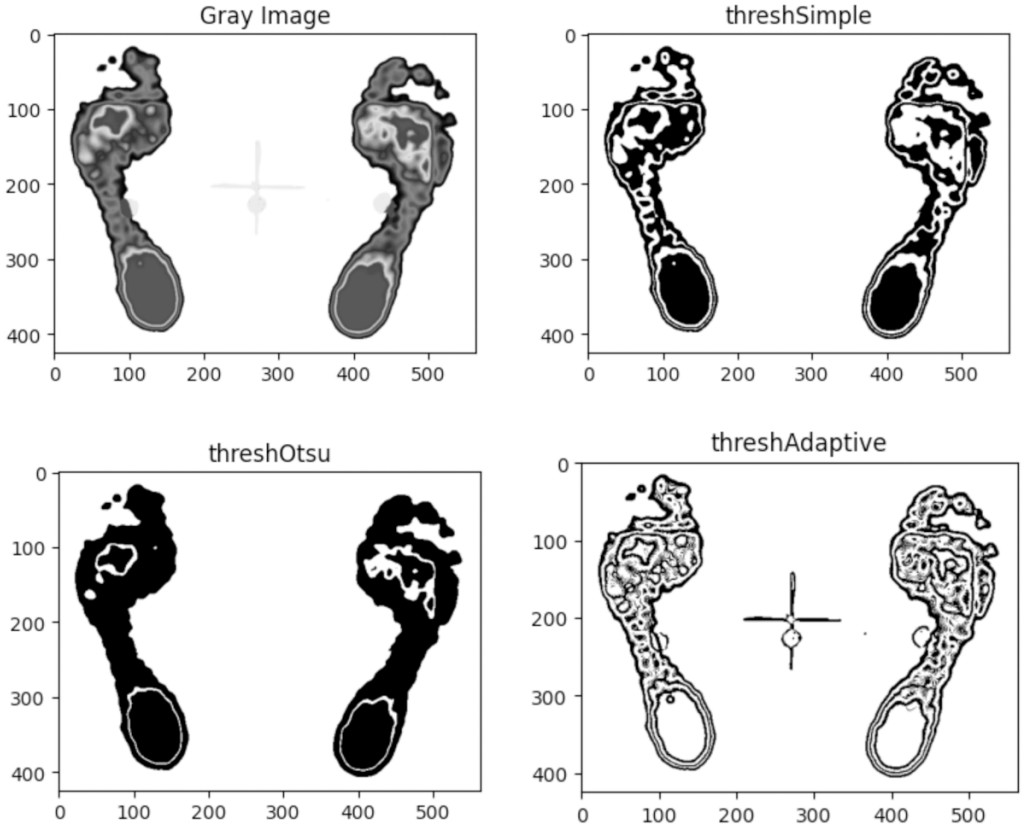

**Figure 5** Comparison of segmentation results for the same plantar image based on three threshold methods: simple, Otsu's, and adaptive.

## Comparison of image segmentation algorithms

Using this test dataset, we compared the performance of three widely used algorithms: thresholding, region split and merge, and Canny edge detection. Each algorithm was evaluated based on its effectiveness in segmenting the forefoot and heel regions.

### Comparison of threshold segmentation methods

Three common threshold segmentation methods (Simple, Otsu's binarization, and Adaptive) were applied to the same image, with results depicted in Fig. 5.

As illustrated in Fig. 5, the adaptive thresholding method captures more detailed small regions, while Otsu's method results in fewer regions. Although the simple threshold method yields less detail than the adaptive method, it successfully outlines the overall contours of the forefoot and heel, which is essential for plantar pressure analysis. Therefore, simple thresholding is recommended as the threshold segmentation algorithm for plantar image segmentation.
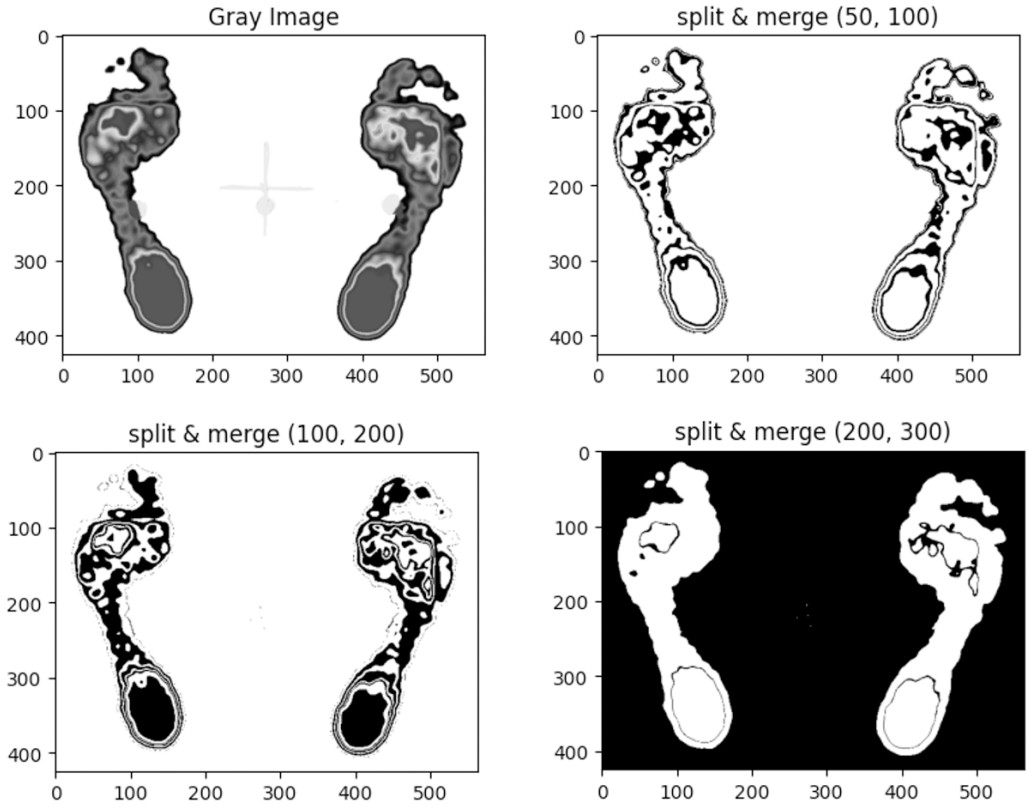

**Figure 6** Comparison of segmentation results for the same plantar image using the region split and merge method with three different merging thresholds.

### Comparison of merging thresholds in split and merge algorithm

To identify suitable merging thresholds (minThresh, maxThresh), we tested three different pairs and compared their segmentation results using the region split and merge algorithm, as illustrated in Fig. 6.

The results indicate that lower threshold values (50, 100) yield more detailed regions, while higher values (200, 300) result in fewer merged regions. The pair (50, 100) was preferred for capturing clearer contours of the forefoot and heel.

### Comparison of double thresholds in Canny edge detection

Three sets of double thresholds (minVal, maxVal) were evaluated for the Canny edge detection algorithm, with results shown in Fig. 7.

The comparison reveals that lower double threshold values (100, 200) produce more refined regions but may lead to discontinuous edges, while threshold values (200, 300) provide fewer segmented regions, with some broken edges. Higher thresholds (300, 600) capture only the outer contour edges, making it suitable for detecting the external contours of the foot.

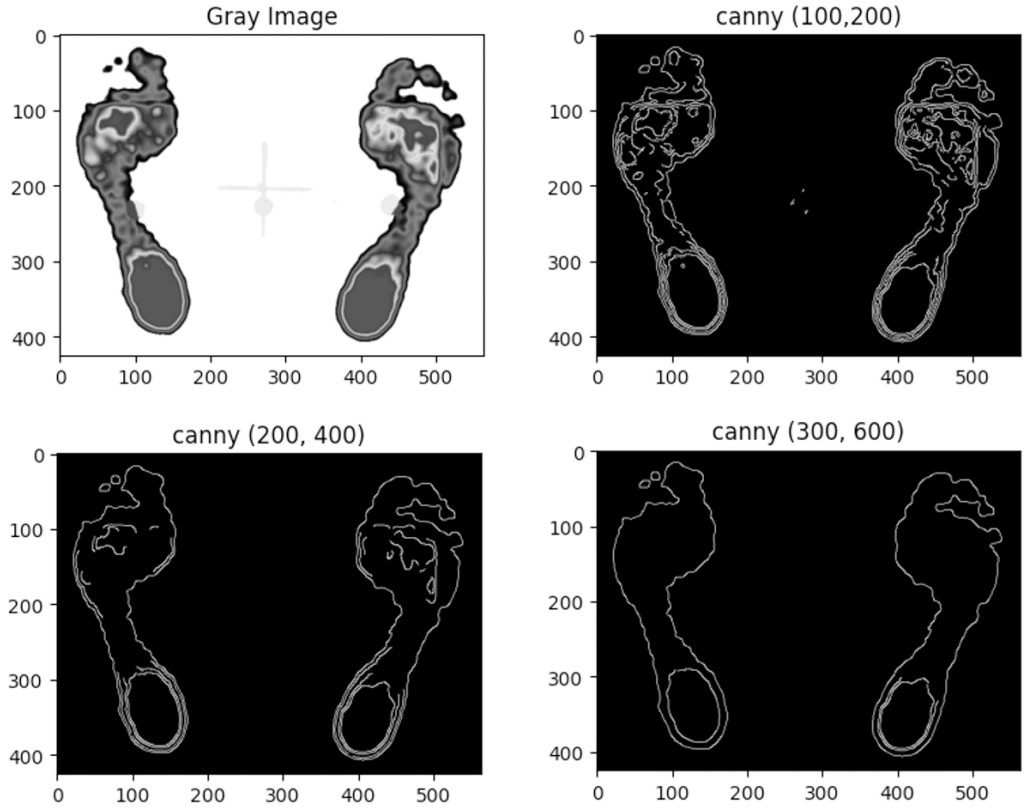

**Figure 7** Comparison of segmentation results for the same plantar image using the Canny algorithm with three different double thresholds.

## Combining filtering contours with geometric method for foot structural segmentation

To enhance contour clarity, we initially applied a filtering approach to three tested algorithms: Simple Thresholding, Split & Merge (50, 100), and Canny (300, 600). The experimental results indicated that while filtering improved contour clarity by removing smaller, less relevant regions, none of the algorithms adequately captured the structural boundaries of the foot (forefoot, midfoot, heel) based solely on pixel connectivity. This limitation motivated the development of our hybrid algorithm.

This section evaluates the effectiveness of integrating geometric structural segmentation with the selected contour and edge detection algorithms. We utilized a test dataset of 200 images, representing a diverse range of normal and abnormal foot types, ensuring that our comparisons are based on a comprehensive dataset reflecting various foot conditions. The performance of the combined geometric method with the three algorithms was rigorously assessed using average Intersection over Union (IoU) and mean Average Precision (mAP) metrics. The IoU and mAP values were calculated in two steps: first, by averaging the results for normal and abnormal foot images to compare differences, and then by averaging these values for the final results.

**Table 1  Performance comparison of the combined geometric method with three different algorithms using IoU and mAP metrics.**

| Methods/Regions | | Threshold with geometric | | Split&Merge with geometric | | Canny with geometric | |
|---|---|---|---|---|---|---|---|
| | | IoU | mAP | IoU | mAP | IoU | mAP |
| Left | outerForefoot | 0.51 | 0.46 | 0.61 | 0.59 | 0.92 | 0.92 |
| | innerForefoot | 0.50 | 0.45 | 0.63 | 0.61 | 0.92 | 0.92 |
| | midfoot | 0.63 | 0.62 | 0.58 | 0.58 | 0.96 | 0.96 |
| | heel | 0.43 | 0.39 | 0.54 | 0.52 | 0.97 | 0.97 |
| Right | innerForefoot | 0.47 | 0.42 | 0.52 | 0.48 | 0.98 | 0.98 |
| | outerForefoot | 0.52 | 0.48 | 0.55 | 0.52 | 0.97 | 0.97 |
| | midfoot | 0.58 | 0.56 | 0.54 | 0.52 | 0.94 | 0.94 |
| | heel | 0.49 | 0.45 | 0.56 | 0.55 | 0.95 | 0.95 |

The mAP is a comprehensive metric for evaluating object detection models, as it considers both precision and recall, allowing for a nuanced assessment of model performance. IoU measures the similarity between predicted and ground truth bounding boxes, calculated as the ratio of their intersection to their union, as expressed in Eq. (13). The ground truth was obtained manually using the Make Sense platform (https://www.makesense.ai/).

$$IoU(A, B) = \frac{(A \cap B)}{(A \cup B)} \tag{13}$$

where A and B represent the predicted and ground truth bounding boxes, respectively. An IoU score of 1 indicates perfect overlap, while a score of 0 indicates no overlap. In this study, algorithm performance was classified as: poor (IoU <0.6), good ($0.6 \leq$ IoU <0.90), and excellent (IoU $\geq$ 0.90).

The performance evaluation results, measured by IoU and mAP, are summarized in Table 1. The hybrid Canny + Geometric method consistently outperformed the other algorithms, achieving IoU and mAP scores exceeding 0.90 for all regions (forefoot, midfoot, heel). In contrast, the other methods exhibited greater variability, particularly in the heel region. This discrepancy in performance can be attributed to the Threshold and Split & Merge algorithms, which focus on detecting contours within the foot's interior, thereby compromising stable detection of the outer edge and leading to inconsistent segmentation. Overall, our findings indicate a positive correlation between higher IoU values and mAP scores, highlighting the effectiveness of the hybrid Canny + Geometric method for accurately segmenting the foot into forefoot, midfoot, and heel regions.

Furthermore, the results demonstrate that the performance of the proposed hybrid algorithm is comparable for both normal and abnormal foot types, with no significant differences observed. The segmentation results for the five foot types using the combined Canny and geometric structural method are illustrated in Fig. 8.

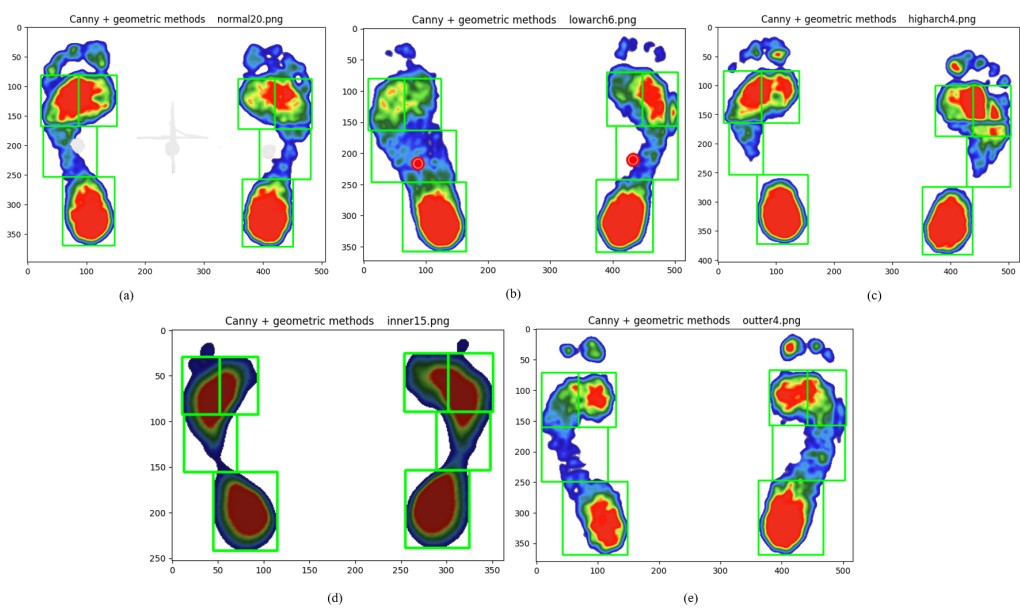

**Figure 8** **Plantar foot region segmentation using the integrated Canny and geometric method for five foot types.** (A) normal, (B) low arches, (C) high arches, (D) inward heel tilt, (E) outward heel tilt.

## DISCUSSION

This study proposes a hybrid contour-geometric algorithm that effectively addresses key challenges in the segmentation of the plantar foot region. Our approach demonstrates high accuracy (IoU/mAP > 0.90) in delineating the forefoot, midfoot, and heel, even across diverse foot morphologies (Fig. 8). This success is achieved by synergistically combining robust edge detection (Canny) with standardized geometric partitioning. The algorithm utilizes edge contours to determine foot length and position, and subsequently applies geometric proportions to define the regions.

The evaluation of three image segmentation algorithms revealed that the combination of the Canny edge detection algorithm with geometric partitioning methods yields the most effective and precise results for segmenting foot regions, particularly in accurately identifying the outer edge of the foot. Our findings align with those of *Boudraa et al. (2024)*, indicating that Canny edge detection provides accurate edge localization and demonstrates robustness against noise. This precise boundary detection is crucial for subsequent analyses of plantar pressure distribution characteristics, which are essential for assessing foot health and designing customized footwear solutions.

While this study advances plantar foot region segmentation, certain limitations must be acknowledged. First, regarding morphological constraints, although the algorithm performs well for standard adult foot morphologies, its accuracy may decline with extreme foot shapes and sizes, and it has not been validated for pediatric populations. Second, concerning pathological limitations, incomplete footprints from conditions like partial forefoot/heel absence may compromise segmentation accuracy. Lastly, in terms

of validation scope, the approach was validated only under static conditions and has not been tested with dynamic footprints (*e.g.*, during gait). Future research should address these challenges through adaptive modeling, expanded pathological datasets, and the incorporation of dynamic datasets.

## CONCLUSIONS

In summary, this study developed a hybrid algorithm that integrates the Canny edge detection method with geometric partitioning to accurately segment plantar regions from static plantar pressure images. The method demonstrates high segmentation precision, providing a robust foundation for subsequent analyses of plantar pressure distribution patterns, which holds considerable potential for informing future foot health assessments and clinical interventions.

## ACKNOWLEDGEMENTS

We would like to express our sincere gratitude to our research team members for their dedication and hard work throughout the project. We also appreciate the support and insights provided by our colleagues in the fields of biomechanics and healthcare, which have greatly enriched our understanding of foot health.

### Funding
This work was supported by the Science and Technology Special Commissioner of Fujian Province, China (Grant No. 202435030011). The funders had no role in study design, data collection and analysis, decision to publish, or preparation of the manuscript.

### Grant Disclosures
The following grant information was disclosed by the authors:
Science and Technology Special Commissioner of Fujian Province, China: 202435030011.

### Competing Interests
Xi Liang and Weiming Gu are employed by Shuangchi Technology Co., Ltd.

### Author Contributions
- Shumei Zhang conceived and designed the experiments, performed the experiments, analyzed the data, prepared figures and/or tables, authored or reviewed drafts of the article, and approved the final draft.
- Xi Liang conceived and designed the experiments, performed the experiments, analyzed the data, prepared figures and/or tables, authored or reviewed drafts of the article, and approved the final draft.
- Minmin Wu performed the experiments, authored or reviewed drafts of the article, and approved the final draft.
- Weiming Gu performed the experiments, authored or reviewed drafts of the article, and approved the final draft.

## Human Ethics

The following information was supplied relating to ethical approvals (*i.e.*, approving body and any reference numbers):

The experimental paradigm was approved by the Research Ethics Committee (REC) of New Engineering Industry College, Putian University (NO. 2023030206), in compliance with the Declaration of Helsinki, written informed consent was waived by the REC.

## Data Availability

The data and code are available in the Supplemental Files.

The data is available at Zenodo, Zhang, S., & Liang, X. (2025). Plantar pressure dataset for foot health assessment (V1.0) [Data set]. Zenodo. https://doi.org/10.5281/zenodo.14621923.

## Supplemental Information

Supplemental information for this article can be found online at http://dx.doi.org/10.7717/peerj.20352#supplemental-information.

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
