# Peer review of "Hybrid contour and geometric partitioning for accurate plantar foot region segmentation"

_PeerJ, doi:10.7717/peerj.20352_

## Round 0.1 · original submission · Major Revisions

· Academic Editor

Major Revisions

**Language Note:** The review process has identified that the English language must be improved. PeerJ can provide language editing services - please contact us at [email protected] for pricing (be sure to provide your manuscript number and title). Alternatively, you should make your own arrangements to improve the language quality and provide details in your response letter. – PeerJ Staff

Reviewer 1 ·

Basic reporting

Literature references, sufficient field background/context provided.

Experimental design

Methods are described with sufficient detail & information to replicate.

Validity of the findings

Conclusions are well stated, linked to the original research question & limited to supporting results.

Additional comments

I appreciate your great work. I found the paper to be well written overall, and much of it to be well described. But I have some comments:

The title is important in a manuscript; the title is not understandable. The authors must rephrase or rewrite the title.

Keywords are not mentioned in the abstract. Use Mesh terminology.

Introduction
Kindly end the introduction in a way that will attract the interest of the readers towards your study.

Study subjects and methods
1. There is a detailed methodology, but it can be concise by adding images, tables, and figures
2. How was the sample size calculated?
Please mention the equation or method used to calculate the sample size
3. Also, elaborate in detail about how individuals were recruited, demographic details
4. Also mention the descriptive statistics of demographic variables like age, gender, and add a table with details of demographic variables

Discussion:
1. Poorly structured statements, no logical sequence of ideas
2. You should state each objective or your result in the first line of the paragraph, and discuss and interpret similar and non-aligned studies

The conclusion should be short, mention future recommendations in the discussion itself, if required, and mention in one line rather than explaining in detail in the conclusion.

·

Basic reporting

The paper deals with a very interesting, though widely discussed, topic, addressed by using quite an innovative approach. Readability is good, and only few typos have been noted by this reviewer.
However, some important concerns raised during the review of the paper, which deserve attention and, possibly, some major interventions.
First concern deals with the too many issues treated, most of which too much superficially. The result is a too long but superficial paper. Thus, a useful suggestion might be to better focus on few relevant issues, and to clearly state the boundary of applicability of the proposed segmentation approach (just to mention: complete footprints are needed; footprints shall be collected under specific test conditions; data collection devices shall prove to have peculiar features,...). Also, mentioning and commenting other approaches to foot segmentation, either in the Introduction or in the Discussion section, might enhance the relevance of the paper.

Experimental design

Data collection methods should be reviewed, so as to allow the reader to fully understand and eventually reproduce the experimental conditions. Among the relevant methodological details (please consider this as an example and not an exhaustive list): volunteers’ main characteristics; data collection protocol; number of footprints per foot and per volunteer; criteria used for foot type stratification; definition of the three foot regions (according to literature); software and other computational instruments used.
Additionally, the whole section/paragraph associated with the computation of pressure distribution raises serious concerns about correctness, validity, proper definition of physical quantities, measurement units and metrics. It is possible that this concern is only due to missing details and unclear explanations. In any case, the suggestion here is to fully remove the issue. Similarly, and for the same reasons, it is recommended to remove the whole attempt to characterise percentages of load for different foot types: it is. or it seems, a pure qualitative dissertation, whose validation would deserve a full study (it may become the key issue of future papers, in case).

Validity of the findings

Validation/comparison of methods: all the paragraphs reporting on picture qualitative observation do not refer to a rigorous validation process, thus the suggestion is to shorten them as much as possible and to eventually leave one or two sentences to comment the observations (concurrently, the number of figures might be reduced as well).
Only rigorously measured comparisons should remain (namely, the IoU and mAP metrics), and for them further information is needed, since even from the supplemental material it is not clear how many footprints have been used for the comparisons, from how many volunteers, and so on (in the text, Authors mention 100 footprints, but the supplemental material report metrics for much less footprints, at least it seems so)
As a last recommendation, the Limitations section should be expanded. Several issues should be reported and commented here, among which: troubles with incomplete footprints due to foot pathologies; impossibility of hallux detection; difficulty to address footprints from seriously deformed feet; applicability to children feet;...

Additional comments

Some typos have been noted, especially in the pseudo-code lines. In general, the overall use of this pseudo-code might be reviewed to ensure paper readability.

The title might be re-formulated, so as to better represent the main aim of the paper

---

## Round 0.2 · Major Revisions

· Academic Editor

Major Revisions

**Language Note:** When you prepare your next revision, please either (i) have a colleague who is proficient in English and familiar with the subject matter review your manuscript, or (ii) contact a professional editing service to review your manuscript. PeerJ can provide language editing services - you can contact us at [email protected] for pricing (be sure to provide your manuscript number and title). – PeerJ Staff

·

Basic reporting

The manuscript has been greatly improved, and I thank the authors for having accounted for most of the comments.

However, some major and minor questions still need to be addressed. I hope the following explanation may help to clarify the request.

The main concern is on the main aim, relevance, and innovation of the study. More in detail: as currently reported in the Abstract, “We propose a hybrid algorithm that integrates edge contour detection with geometric partitioning to enhance region-specific segmentation accuracy”. In which sense? Most commercial software packages associated with plantar pressure measurement devices indeed already exploit the footprint features, among which is its contour, before applying the geometry-based regionalization of the foot. If you apply the same region definition – same criteria and same proportions – to a pressure footprint after having identified the contour according to your methodology and to the same footprint through the commercial software package of the measurement device, you get exactly the same result. And you surely know this very well. On the other side, if, after having identified the contour, you apply a simple geometry-based criterion to define the foot regions, what are you offering more than the existing procedures/software? In what sense can you enhance the accuracy of the region detection? Thus, it is obvious that there is something in your aim which is not very clearly expressed or declared, yet. In other words, though very interesting, as it is presented right now, commercial pressure measurement devices and associated software packages do not need this contour identification. Please think about this and try to clarify the issue.

Experimental design

On the crucial issue of reference papers, relevant citations are missing at several points, among which: the definition of the geometric proportions and criteria for the foot regions, the definition of foot types, and the published methods for foot pressure partitioning.

In the methods: if you acquire two feet from 400 individuals, you get 800 footprints, not 400. Can you please explain better? Did you use only one foot per person? Which one, in case?

In the methods, was the position on the platform fixed or free?

In the methods, you mention you acquired 15 seconds per person: which resulting footprint did you analyse? An average one? An instantaneous one? Other?

In the methods, you mention you selected 100 out of 400 footprints: why? With which criterion?

Validity of the findings

Limitations, though appreciated, should be further reviewed. Or, rather, the very limited conditions for the validity of the proposed approach should be more clearly stated, for example, the fact that it has neither been tested with dynamic footprints nor against gold-standard methods for pressure footprint regionalization.

Additional comments

Appropriate terminology is important. What you report as “arch” region is mostly identified as “midfoot” region; you can check the reference literature to verify this observation.

---

## Round 0.3 · Minor Revisions

· Academic Editor

Minor Revisions

·

Basic reporting

The Authors were committed to addressing all previous comments in the revised manuscript, and this was indeed appreciated.

A crucial concern remains for this Reviewer, which, at this point, seems to be due to the Reviewer's limitation in clearly explaining it.

It is briefly reported once again below, leaving to the Editor the final decision whether to still ask for clarification or to simply ignore it.

Briefly, what is reported in the Conclusions of the manuscript, both in the Abstract and in the Manuscript Conclusions, is not formally true and should not be accepted. To further clarify, the sentence in the Abstract Conclusions, "This approach significantly enhances the precision of anatomical region identification and offers a robust foundation for analyzing plantar pressure distribution across diverse foot morphologies," cannot be accepted: in which part of the manuscript do the Authors prove this? What the Authors did, and reported in the manuscript, was to compare the performance of their algorithm ONLY IN CASE OF STATIC STANDING PRESSURE FOOTPRINTS and ONLY WITH RESPECT TO THREE OTHER IMAGE SEGMENTATION ALGORITHMS. Honestly, the mentioned "Anatomical region identification" refers to a much wider field of application, and the thorough validation of a method with respect to it should go much further than the study reported in the manuscript.

Experimental design

-

Validity of the findings

-

---

## Round 0.4 · accepted · Accept

· Academic Editor

Accept

Thank you for addressing the reviewer comments. Your manuscript is now considered suitable for publication.

·

Basic reporting

Reviewer's comments addressed

Experimental design

Reviewer's comments addressed

Validity of the findings

Reviewer's comments addressed